# Numerical Modelling Assisted Design of a Compact Ultrafiltration (UF) Flat Sheet Membrane Module

**DOI:** 10.3390/membranes11010054

**Published:** 2021-01-14

**Authors:** Mokgadi F Bopape, Tim Van Geel, Abhishek Dutta, Bart Van der Bruggen, Maurice Stephen Onyango

**Affiliations:** 1Department of Chemical Engineering, KU Leuven, Celestijnenlaan 200F, B-3001 Leuven, Belgium; bart.vanderbruggen@kuleuven.be; 2Department of Chemical, Metallurgical and Materials Engineering, Tshwane University of Technology (TUT), Private Bag X680, Pretoria 0001, South Africa; OnyangoMS@tut.ac.za; 3Department of Chemical Engineering, Izmir Institute of Technology, Gülbahçe Campus, Urla, Izmir 35430, Turkey; abhishekdutta@iyte.edu.tr

**Keywords:** ultra-low pressure (ULP), ultrafiltration (UF), simulation, computational fluid dynamics (CFD), flat sheet membrane module

## Abstract

The increasing adoption of ultra-low pressure (ULP) membrane systems for drinking water treatment in small rural communities is currently hindered by a limited number of studies on module design. Detailed knowledge on both intrinsic membrane transport properties and fluid hydrodynamics within the module is essential in understanding ULP performance prediction, mass transfer analysis for scaling-up between lab-scale and industrial scale research. In comparison to hollow fiber membranes, flat sheet membranes present certain advantages such as simple manufacture, sheet replacement for cleaning, moderate packing density and low to moderate energy usage. In the present case study, a numerical model using computational fluid dynamics (CFD) of a novel custom flat sheet membrane module has been designed in 3D to predict fluid flow conditions. The permeate flux through the membrane decreased with an increase in spacer curviness from 2.81 L/m^2^h for no (0%) curviness to 2.73 L/m^2^h for full (100%) curviness. A parametric analysis on configuration variables was carried out to determine the optimum design variables and no significant influence of spacer inflow or outflow thickness on the fluid flow were observed. The numerical model provides the necessary information on the role of geometrical and operating parameters for fabricating a module prototype where access to technical expertise is limited.

## 1. Introduction

Access to drinking water is a basic human right and providing access has become an international goal as envisaged in the Sustainable Development Goals (SDGs) [1,2]. Within the South African context, the percentage of the population using safely managed drinking water services has remained constant at around 91% between 2014 and 2016 as reported in the South African SDG baseline report [3]. Approximately 26% of the population in the Mutale local municipality (Limpopo province, Venda) have poor access to municipal services (piped drinking water), leading to relatively bad access to clean drinking water [4,5]. Tsaande B, also known as Tshaanda village, is located in the Mutale Local Municipality (GPS coordinates: 22.6857 S, 30.4189 E) within the Limpopo province, South Africa [6]. With aims to combat issues related to poor water quality and supply and consequently, failure of centralized reliable water distribution networks, a pilot-scale decentralized Ultrafiltration (UF) membrane plant was installed for provision of drinking water. The plant comprised a commercial hollow fibre UF membrane (Multibore membrane housed in a dizzer XL 1.5 MB 40) for removal of bacteria, viruses, suspended particles and colloids. Moreover, the ultimate goal was to demonstrate the feasibility of membrane technology to provide drinking water in a rural area [6]. The system operation started in 2014 and although the UF pilot plant could produce safe drinking water, much work was still required before this technology could be rolled-out countrywide. Successive research expansion was to include evidence of sustained use on water quality over extended periods, operation and maintenance, cost-effectiveness, system performance and optimization (considering seasonal variations), and water quality monitoring [6]. Up until now, no follow-up evaluation has been made. Furthermore, the dependence of the system on a commercial membrane module presented opportunities to develop a custom membrane module producible on a local scale, to lower capital cost requirements.

The development of membrane module design, fabrication and optimization on operating conditions is built on a wide range of prior arts and practical experiences. In general, the criteria for membrane module configurations include two types: namely, flat sheet and tubular membrane modules [7]. Numerous studies [8,9,10,11,12] have explored the design of the hollow-fibre (tubular) configuration [13], spiral wound [14] due to competitive benefits; however, the design and fabrication of hollow fibre modules is a complicated process that involves different disciplines and requires a thorough understanding of the intended application, particularly for decentralized water treatment systems [13,15]. Nonetheless, a study by Oka et al. [16] explored the operation of a submerged hollow-fibre UF system for drinking water for application in small/remote communities. However, the complexity of the system on auxiliary fouling control measures including backwashing, air sparging and chemical cleaning prove to be costly. Furthermore, three different configurations of bench-scale commercial submerged hollow-fibre membrane systems were used, adding more cost to the overall costs of the system. Pillay et al. [17] presented a technical report on the development of a capillary ultrafiltration (CUF) system for a small-scale potable water treatment system for rural and peri-urban areas. Periodically, the information from the technical evaluations was combined with the information of sustainability needs. Once again, a commercial membrane module was used. Consequently, a guidebook on the selection of small water treatment systems for drinking water supply to small communities was published [18]. Although the merits and drawbacks of the different configurations can be argued, the plate and frame configuration may offer certain advantages over other module configurations, particularly for decentralized membrane water plants in small/remote communities [19,20,21]. Flat sheet plate and frame membranes offer benefits in simplicity, better flow control on both the permeate and feed side [22], ease of sheet replacement, less fouling tendency over tubular configurations (excessive fouling and membrane integrity problems) [7,23], while also being easier to prototype in a laboratory environment.

Computational fluid dynamics (CFD) is a widely integrated tool for process optimisation [24,25] and design of UF water and wastewater treatment systems to achieve a desired system performance [26,27,28,29,30,31,32,33,34,35]. The fundamental characteristics of membrane module design comprises minimizing the cost per amount of mass transferred and maximizing the system performance through optimized flow geometry and operating conditions [9]. Moreover, other aspects in water treatment plant design may include: (1) meeting water quality standards and requirements; (2) minimizing overall project costs; and (3) controlling ongoing operating costs and maintenance requirements. Successfully achieving these objectives is dependent upon proper design and optimization of the treatment water flow systems within the plant. Numerous studies [35,36,37,38,39,40] have employed CFD tools to gain an insight into the phenomena taking place inside membrane modules [33,41,42,43,44] and to improve the overall performance of modules. As an analysis tool, CFD provides the ability to modify operating conditions, fluid properties and geometric characteristics of the flow channels in a flexible but defined way [31]. For example [45], the presented results on the design of a thin channel cross-flow module were for the characterization of flat ceramic membranes, with a primary objective ensuring uniform flow characteristics over the permeating area. Boundary conditions were imposed such that the flow non-uniformity was taken as the normalized standard deviation of the velocity field above the permeating area, while the pressure drops considered were those across the inlet plenum and across the permeating area normalized with respect to the outlet pressure. Similarly [46], the presented results of CFD modelling of flow within the SEPA CF flat sheet membrane filtration cell operated at low recoveries. As expected, the goal of the study was to characterize uniform flow distribution within the cell. The common Navier–Stokes governing equation was applied, with all the walls set at a no-slip boundary including the membrane. Results revealed stagnation areas in dead ends of the inlet and outlet tubes and in the channel areas behind duct entries as well as local regions of high shear in duct-channel transition areas (the flow was unidirectional over most of the channel area with exception of the corners of the channel). Another example was presented by [33], who performed CFD studies on a circular cross-flow NF laboratory test cell with an aim to improve the design of the cell and to increase the filtration performance. Design variables considered were the feed chamber thickness, number and distribution of the inlet/outlet pipes, sequential mode inlets and the addition of grooves on the top surface of the cell. The two dimensional (2D) study demonstrated that the groove size and the groove interval play a significant role. Therefore, for small bench-scale cells when the use of retentate spacers is not convenient, the alternative would be to add grooves opposite to the membrane side in order to create velocity fluctuations [33,47]. In general, a membrane performance is significantly influenced by the operating pressure and velocity within a membrane module; therefore, it is crucial to obtain and maintain uniform fluid flow hydrodynamic (pressure and velocity) conditions over the membrane surface [10,16]. Commercially available codes are well suited for the types of simulations required to advance water and wastewater component system designs; however, each code has individual strengths and weaknesses to be considered.

To characterize the performance of a custom membrane module, this study evaluated the design, development, and modelling of fluid flow through a flat sheet plate and frame module prototype, using CFD techniques, to identify the geometry (spacer geometry, spacer thickness, inlet and outlet diameter, inlet and outlet length) and membrane parameters (permeability, membrane thickness, membrane area) that may influence the module’s overall performance. Moreover, the aim of the study was to design a membrane module with uniform flow characteristics over the permeating area. Da costa et al. [48] and Saeed et al. [43] acknowledged that the spacer geometry is critical in module performance as it directly relates the hydrodynamics and solute transport. Therefore, the type of the spacer used will strongly influence the resulting flow and ultimately, the performance of the module [48,49,50,51]. The development of a model that can be used to simulate the module prototype may give an insight [52] into whether lab-scale performance corresponds with pilot scale performance, reducing the need for experiments during its optimization process and effectively reducing optimization costs. Subsequently, the model can be validated using experimental data.

## 2. Materials and Methods

### 2.1. Proposed Module Design and Fabrication

#### Design Dimensions and Variables

The design and development of the proposed module prototype in the present study was achieved using AutoCAD 2D as a drawing tool. Additionally, the design was based on local process conditions collected but flexible enough to be applied in a different region. The module housing consisted of two Perspex glass closing holders (thickness 20 mm), five tortuous spacers (thickness 2 mm), used for flow channelling and sealing of the stack with corresponding gaskets (thickness 2 mm). The thickness of the spacers was based on the required flow velocity on the membrane surface and their quantity was based on the membrane flux. The overall dimensions of the unit were 20 cm × 20 cm × 25 cm, while the effective membrane surface was 0.01 m^2^. The significance of the tortuous (plug-flow) permeate spacer design was to bring about high flow velocity over the membrane (membranes arranged horizontally), to enable cleaning of the membrane surface while under operation and support the membrane sheets mechanically. No mesh in the feed side was used to avoid the possibility of heavier total suspended solid load accumulating in the feed side. The inlet and outlet ducts were configured by etching the Perspex holder (etching depth of 1.5 mm). Valves and fittings are also important for the stack operation and thus they were designed in such a way that they are quick to connect and disconnect from tubes and completely waterproof, and most importantly, provide resistance against corrosion. Three speed controllers as quick connectors to outlet diameters were also included, to control flow and then lock to sustain the desired flow.

### 2.2. Development of the CFD Modelling

Numerical simulation platform ANSYS^®^ 19.2 (ANSYS Inc., Canonsburg, PA, USA) was used to model and simulate the membrane module. The geometry was designed using an ANSYS Design Modeler, while ANSYS Meshing was used to create the required grids for pre-processing. ANSYS Fluent was used to set up and solve the actual model and ANSYS CFD Post was used for post-processing tools. Simulations were carried out on two computer systems: an MSI Aegis X3 desktop computer, with 16 GB of RAM and a quad core 4.2 GHz Intel Core i7-7700K processor, overclocked to 4.5 GHz, and a MSI GE63 Raider laptop computer, with 16 GB of RAM and a hexa core 2.2 GHz Intel Core i7-8750H processor, overclocked to 4.1 GHz. A brief description of the steps followed within the ANSYS workbench GUI to set-up a CFD model, i.e., the creation of a geometry, formation of a grid (mesh), and set-up of the model by combining different models for hydrodynamics and turbulence is presented.

#### 2.2.1. Geometry and Grid Creation

The geometry simulated in this study is illustrated in Figure 1. A three-dimensional (3-D) mathematical model was developed by reversing the set-up of the geometry (technical drawings), i.e., the flow path was constructed in ANSYS Design Modeler, not on the surrounding materials.

Three distinct zones in the geometry were created: an inflow zone, outlet zone and membrane zone. The inflow zone represents the inlet of the module connected to the flow path through a spacer and a retentate outlet, the outflow zone, found on the other side of the membrane zone and a membrane zone, found at the other side of the spacer when compared to the inlet as shown in Figure 2.

Given the sharp corners of the spacer, the spacer “curviness” was used as a measure of the roundedness of the corners of the flow field through the spacer and calculated using Equation (1) as demonstrated in Figure 3.
(1)$$C=\frac{r_{qc}}{r_{qc,\;max}}$$
where $r_{qc}$ is the radius of a quarter-circle replacing the corners of the flow path and $r_{qc,max}$ is the maximal value possible for this radius within the width of the flow channels.

The quality of a CFD solution is highly dependent on the quality of the grid (mesh). Several grids were created to run the computational domain and each grid had structured multigrid of hexahedral [53,54] cells, to reduce computational time and thus speed up convergence. In all simulation cases, the outflow zone was coarsely meshed as illustrated in Figure 4, while the membrane and the spacer zone were finely meshed, as shown in Figure 5.

To test the quality of the size of the grid cells on the simulation solution, a grid analysis was carried out. Five grids were generated, varying from coarse to fine, with 250,000, 500,000, 1,000,000, 1,500,000, and 2,000,000 cells. Based on the assumption that finer meshes typically lead to an accurate solution, but with higher computational costs, a balance between computational costs and solution accuracy is paramount. Therefore, a parametric sensitivity analysis on the geometrical parameters was conducted.

#### 2.2.2. Governing Equations

The fluid was assumed to be isothermal, incompressible, and Newtonian and a steady-state flow was formulated in the three-dimensional (3D) domain. The modelling was based on a single-phase flow and pure water flux was modelled through the module. The simulation was implemented by solving hydrodynamic Navier–Stokes governing equations, including mass and momentum conservation equations. In general, the mass and momentum conservation equations [35,55] can be expressed as follows:(2)$$\frac{\partial_{p}}{\partial_{t}}+\nabla \cdot \left(\rho v\right)=0$$
(3)$$\frac{\partial \left(\rho v\right)}{\partial_{t}}+\nabla \cdot \left(\rho v\mu \right)=-\frac{\partial P}{\partial_{y}}+\nabla \cdot \left(\mu \nabla v\right)+SM_{y}$$
where ρ is the density of the fluid (kg/ m^3^), *v* is the component of the fluid velocity in the x direction (m/s), *µ* is the fluid viscosity (Pa s), *P* is the pressure (Pa), *t* is time (s), and *SM_y_* is a source term in the y direction (kg/ m^2^ s^2^). For constant density fluids, the continuity equation states that the velocity field is non-divergent [31] as expressed in Equation (4).
(4)∇·v=0

#### 2.2.3. Solution Method

The commercial package ANSYS^®^ 19.2 (ANSYS Inc., Canonsburg, PA, USA), based on a finite element method (FEM), was used to simulate the laminar flow in the module prototype. At the inlet, a normal inflow velocity boundary condition (i.e., imposed uniform velocity profile) was given. At the outlet, a zero-gauge pressure was set since fluid flows out of the computational domain. At all wall boundaries, even at the membrane surface on the bottom of the cell, the no-slip condition was imposed. The expressions were discretized by the Semi-Implicit Method for Pressure-Linked Equations (SIMPLE) algorithm [56], coupled with velocity to pressure in the system to solve the flow field. An iterative geometric multigrid (coarse, medium, and fine) solver was chosen. The membrane was defined as a porous zone and the pure water flux through the membrane was defined by Darcy’s Law [57], calculated using Equation (5).
(5)$$J=\frac{k}{\mu }\frac{\Delta p}{\Delta x}$$
where *J* is the flux (L/m^2^·h), *k* is the intrinsic permeability of the membrane (m^2^), ∆*p* is the transmembrane pressure (TMP) over the membrane (Pa), *µ* is the dynamic viscosity of the fluid passing through the membrane (Pa·s) and ∆*x* is the thickness of the membrane (m). Consequently, the pores were not modelled individually, instead the membrane was regarded as a black box with a value for the intrinsic permeability inserted into Darcy’s law. The intrinsic permeability for the reference case was chosen as 4.167 × 10^−17^ m^2^, which corresponds to a permeability of 100 L/m^2^.h.bar, when the thickness of the membrane is 150 µm and the water viscosity is 1 × 10^−3^ Pa·s, as calculated using Equation (6),
(6)$$K=\frac{k}{\mu \Delta x}$$
where *K* is the permeance, *k* is the intrinsic permeability (m^2^), *µ* is the dynamic viscosity (Pa·s) and ∆*x* the membrane thickness (m^2^). If a membrane with this permeability value is used at a constant transmembrane pressure of 20,000 Pa (0.20 bar), a constant flux of 20 L/m^2^h can be achieved. To achieve a water flux of 5000 L of clean water per day, ca. 260 membranes of 20 cm by 20 cm are required. However, this value assumes a constant TMP, for which it cannot be achieved in a crossflow system with a spacer due to pressure drop and a complete use of the membrane area due to the spacer. To validate the performance of the model, the intrinsic permeability k, the transmembrane pressure (TMP) ΔP, the inlet pressure and the membrane thickness Δx were varied in a sensitivity analysis presented in Table 1. During simulation of these parameters, only pure water flux was considered, and no fouling or concentration polarization phenomena were considered in the model.

## 3. Results and Discussion

### 3.1. Grid Sensitivity Analysis

To analyze the influence of the grid on the simulation solution quality, the cell count of the grid was varied.

#### 3.1.1. Grid Cell Count Quality

Generally, it may be assumed that as the number of cells within a grid increases, the quality of the computational solution increases due to the fineness of the grid, because as the grid is refined, the numerical discretization error is reduced and the solution approaches convergence (continuous profile). Figure 6 shows how the transmembrane pressure (TMP) develops throughout the spacer flow path by plotting its value along a centreline, i.e., a line drawn in the middle of the spacer flow path, at 5 mm distance from the spacer walls. The most significant differences between the five grid cells occur very close to the inlet, where pressure and velocity gradients are largest. Evidently, the pressure drops throughout the system, starting from a gauge pressure of 20,000 Pa (equal to the inlet pressure of 0.2 bar) to 0 Pa, close to the retentate outlet. This pressure drop can be related to both the lengthening of the flow path and spacer due to the frictional forces caused by the resistance to flow. Additionally, slight pressure spikes and drops around corners of the flow path defined by the spacer can be seen. This may be regarded as flow “hitting” into the walls of the spacer due to the viscous shear stresses. The pressure drop approaches linear, towards the retentate outlet, where the flow is nearly fully developed, as expected. With an assumption that increasing grid cells improves the simulation resolution (i.e., 2,000,000 grid cells), a slight difference was observed between grids with 250,000 and 500,000 cells on the one hand, and the 2,000,000-cell grid reference case on the other hand. The pressure profiles for both cases lie below the reference case. The 1,000,000 cell grid has a profile lying slightly above the reference profile, while the pressure profile for the grid with 1,500,000 cells almost entirely overlap with the pressure profile for the 2,000,000 cell grid. The difference in pressure drop from the coarse grid to the fine grid is less than 10%. Subsequently, results from the sensitivity analysis study were compared with results found for the grid with 2,000,000 cells, and in choosing the optimal grid for further simulations, a grid with 1,500,000 cells was chosen as a balance between accuracy and computational cost.

#### 3.1.2. Inclusion of an Outflow Zone

The transmembrane pressure (TMP) profile along the centreline within the spacer flow path with 1,500,000 cell grid, both for an outflow zone and no outflow zone is shown in Figure 7. A distinct difference for both cases could not be established, due to the reference and outflow zone pressure profiles overlapping, inferring that TMP has no significant effect when either an outflow zone is included or when it is not, therefore removing the outflow zone could be a valid approach.

The velocity contours for the spacer flow path right above the membrane are shown in Figure 8. As can be seen, the flow distribution remains uniform as the grid is refined and does not differ when either an outflow zone is included or an outflow zone is excluded, validating the assumption of excluding the outflow zone, as it does not significantly affect the simulation solution. However, the mean velocity in the system was found to decrease as the cell grid count increased. Although the pressure drop varied by a few percent as the grid is refined, the flow field remained close to zero near the spacer curviness due to the no-slip condition. Nonetheless, solutions obtained with coarse grids (250,000 and 500,000 cells) are not representative of reality due to poor quality solution.

### 3.2. Parametric Sensitivity Study on Geometrical Variables

Multiple geometrical parameters were varied in a series of simulations, to test the sensitivity of the model to changes in the geometry of the membrane module. The purpose was to establish optimum conditions and configurations to yield a module design with maximum performance. Table 2 lists the parameters.

#### 3.2.1. Spacer Thickness (Inflow Zone) and the Outflow Zone

The influence of the spacer thickness, i.e., the thickness of the inflow zone, on the transmembrane pressure (TMP) profile was varied with different spacer thickness values. As the spacer thickness decreases, the transmembrane pressure develops a smooth profile. When the thickness increases, the pressure drops and peaks around the corners of the spacer. This may be attributed to the thick spacer, as when the fluid flow resistance increases, it leads to high pressure drops, particularly around the spacer corners. The subsequent pressure increase, when the bend in the spacer is rounded, also increases with the thickness of the spacer. To characterize the influence of the spacer inflow or outflow thickness on the fluid flow, the relationship between the permeate flux and reference value is not shown, as no significant effect was found on the simulation solution.

#### 3.2.2. Spacer Curviness

The effect of the spacer curviness was varied through six simulations (i.e., 0%, 20%, 40%, 60%, 80% and 100%) and results are shown in Figure 9. No singularity was found between the pressure field profiles along the centreline within the spacer flow path for different values of curviness (insert image). However, by replacing the “blocky” geometry of the spacer with a more curved geometry, the pressure profile slightly smoothed-out. Schwinge et al. [58] characterized a zigzag spacer for ultrafiltration in spiral wound modules for water treatment applications. The zigzag spacer is thought not to be easily plugged (free from any obstacles) in the flow direction, but the tortuous channel promotes turbulence effects [58]. From the simulation solution, it was observed that when the spacer curviness percentage increased, the pressure drop was less as the pressure profile smoothed-out (stabilized). An increase in the spacer curviness percentage was found to reduce the turbulence-promoting properties of the spacer and may offer localized dead spots with poor mass transfer that encourage fouling. Nonetheless, the zigzag (tortuous) spacer with reasonable curviness may be used in the module prototype to produce a high and stable water quality over a long time frame due to lower associated pressure drop, leading to lower operational costs.

The velocity contour profiles for flow through the spacer inflow path are shown in Figure 10. Although the flow distribution remains uniform in all six cases, the spacer curviness was found to slightly influence the progression (change) of the pressure as the fluid flows through the spacer. As the spacer curviness increases, the flow velocity developed a smoothed-out profile with values of the velocity perpendicular to the flow direction, showing an increase on the wall shear stress and thus the flow evenly distributed over the width of the pipes, especially in the bends of the geometry. This trend also illustrated that the turbulence-promoting properties of the spacer are reduced when the curviness of the spacer geometry increases, which may lead to an onset of fouling and concentration polarization (CP) phenomena when the module is in operation. To study whether this may be the case, mass transfer phenomena investigations are required.

The water flux through the module as a function of the percentage of the reference value for the curviness is shown in Figure 11. To be able to plot the other values of curviness, the percentage of the reference value as shown on the horizontal axis was defined as the value of the curviness subtracted from the 100% value for the reference, i.e., 100% corresponds to 0% curviness, 80% corresponds to 20% curviness, and so on. As can be seen, the permeate flux through the membrane slightly decreases with an increase in spacer curviness, albeit a slight decrease (from 2.81 L/m^2^h for 0% curviness to 2.73 L/m^2^h for 100% curviness). It is well acknowledged that the main parameter for predicting permeate flux is the pressure and velocity upon the membrane [38], where the pressure field depends significantly on the fluid flow pattern. Therefore, the fluid pattern played a significant role in the decrease of the permeate flux with an increased spacer curviness.

#### 3.2.3. Inlet and Outlet Pipe Length

The inlet and the retentate outlet pipe length was varied individually in two case studies consisting of seven simulations each (i.e., 10 mm, 15 mm, 20 mm, 25 mm, 30 mm, 35 mm and 40 mm). The reference value for both lengths was 25 mm. No significant pressure differences were found between the transmembrane pressure (TMP) along the length of the centreline, and therefore these profiles are not provided. A similar observation was also found for the permeate flux and the TMP profiles. Overall, these parameters remained uniform regardless of the inlet pipe or the outlet pipe length. For variation in the inlet length, an average value for the pure water flux through the module, taken over all cases, was 2.822 L/m^2^h, with a standard deviation *(σ)* of 0.010 L/m^2^h. Similarly, the average flux value taken over all cases where the outlet length was varied, was equal to 2.822 L/m^2^h, with *σ* = 0.011 L/m^2^h. Both these values are very small when compared to their corresponding average value (0.34% and 0.41% of the average value for the inlet length variation and the outlet length variation), leading to the assumption that the permeate flux does not depend on either the inlet length or the outlet length. When the TMP was averaged over the results of all cases for the inlet length, a value of 9412.7 Pa was found, with *σ* = 50.9 Pa. The average value for the TMP for the outlet length was 9397.1 Pa, with σ = 49.5 Pa. The standard deviations were 0.54% and 0.53% respectively of their corresponding average values. Similarly, these values are relatively small, and thus it may be assumed that the TMP does not depend on inlet or outlet length.

#### 3.2.4. Inlet and Outlet Pipe Diameter

Two case studies were carried out in which the influence of the inlet and retentate outlet diameter was studied, with seven different values for each respective case study (i.e., 5 mm, 6 mm, 7 mm, 8 mm, 9 mm, 10 mm and 11 mm). Figure 12 shows the pure water flux as a function of the reference value (11 mm). As can be seen, a slight decrease in the permeate flux was found when the retentate outlet diameter increased. When the inlet diameter increases, a slight increase in the permeate flux was observed. This observation can be attributed to that with an increase in the inlet diameter, more water at a hydrostatic pressure of 0.2 bar is allowed to enter the membrane module at once, allowing the inlet pressure to be maintained further along the spacer flow path, and effectively reducing the pressure drop between the inlet and the retentate outlet.

### 3.3. Sensitivity Analysis on Membrane Properties and Operating Conditions

#### 3.3.1. Membrane Permeability

The influence of the intrinsic membrane permeability on the model was tested by carrying out seven different simulations with different values (i.e., 1.042 × 10^−17^ m^2^, 2.048 × 10^−17^ m^2^, 3.125 × 10^−17^ m^2^, 4.167 × 10^−17^ m^2^, 5.209 × 10^−17^ m^2^, 6.251 × 10^−17^ m^2^ and 7.292 × 10^−17^ m^2^). Membrane permeability, membrane thickness and inlet pressure are all important and often interdependent factors. The intrinsic membrane permeability data were compared to the relationship between the total pure water permeate flux leaving the system and the reference inlet pressure of 0.20 bar, and results are shown in Figure 13.

As a first step of the experimental work, the system was operated in a dead-end mode to establish the flow regime and membrane resistance. For this purpose, the outlet pressure was set to be equal to the atmospheric pressure. In order to validate the CFD simulation to predict permeate flux by neglecting the membrane resistance, a linear relationship between the flux and the transmembrane pressure are obtained, which proves that the *Rc* is zero. Otherwise, due to the blocking effect of the cake, a non-linear relationship between the flux and the transmembrane is expected.

#### 3.3.2. Membrane Thickness

The membrane thickness was varied between 50 µm, 100 µm, 150 µm, 200 µm, 250 µm, 300 µm and 350 µm, respectively. The pressure profile development through the spacer zone, the membrane zone and the outflow zone, found in the middle of the centreline, for each of these simulations, is shown in Figure 14. Evidently, the pressure drops throughout the membrane zone. Due to the varying membrane thickness, the slope in the decrease of the pressure in the membrane zone decreases with an increase in thickness. This occurs because transmembrane pressure remains nearly constant for all simulations, while the membrane thickness does not, and since the intrinsic permeability was defined as a constant through the entire membrane zone, (the membrane resistance was neglected), a linear relationship was found over the membrane thickness to obey Darcy’s Law.

#### 3.3.3. Membrane Surface Area

For simulation purposes, it is desirable to have a uniform flow and fluid properties over the permeating area. Six simulations were carried out with membrane squares of 16 cm, 17 cm, 18 cm, 19 cm, 20 cm and 21 cm respectively, corresponding to area sizes of 0.026 m^2^, 0.029 m^2^, 0.032 m^2^, 0.036 m^2^, 0.040 m^2^ and 0.044 m^2^ respectively. The flux declined with an increase in membrane area size as shown in Figure 15, while no significant effects can be distinguished on the transmembrane pressure profile with an increase in the membrane area size. This flux decline may be attributed to that; the effective membrane area is determined by the spacer on the retentate side of the membrane. Therefore, the spacer should approximately be of the same dimensions as the membrane to obtain an optimum flux.

#### 3.3.4. Inlet Pressure

The relationship between the total pure water permeate flux leaving the system and the percentage of the reference value (0.20 bar) was given in Figure 13. As described before, a linear dependency was found between the inlet pressure and the permeate flux. These observations obey the Darcy’s Law, where an increase in transmembrane pressure leads to a proportional increase in the permeate flux. The law was properly added to the model and simulated within the confines of the geometry and the grid.

#### 3.3.5. Overview of the Parametric Sensitivity Analysis

The overall aim of the sensitivity analysis for both the geometry and operating conditions was to define parameters that may influence the module prototype design, relating the flux through the module and the transmembrane pressure profiles (uniform fluid flow conditions). The spacer geometry and inlet diameter played a significant role in obtaining uniform permeating (flow distribution and transmembrane pressure) conditions, ultimately reducing the overall pressure drop. Table 3 lists the operational parameters comprising the inlet pressure, membrane permeability, membrane thickness and area size. All were found to have a significant influence on the permeate flux. A significant influence on the transmembrane pressure profile was found for the inlet pressure, in addition, the spacer thickness and the outlet pipe diameter also showed a significant influence on the transmembrane pressure profile. No singularity was found between the thickness of the outflow zone, the length of the inlet pipe and the length of the outlet pipe on either the permeate flux or the transmembrane pressure, while the membrane area had a significant impact on the transmembrane pressure.

## 4. Conclusions

This study presented the results from a computational fluid dynamics (CFD) numerical study on a compact membrane module prototype intended for decentralized UF drinking water purification, by characterizing the proposed design geometry and predicting the performance of the module prototype. 3D simulations on several geometry parameters were analysed within a parametric and sensitivity study. The main findings were;

The numerical simulation solution could predict permeate flux with a reasonable error. The pressure distribution upon the membrane was found to depend on the fluid flow pattern on the membrane.A parametric analysis on configuration variables was carried out to determine the optimum design variables. The spacer geometry (tortuous spacer) with reasonable curviness was found to impact the permeating conditions and thus, may be used to produce a high and stable water quality over a long-time frame due to the lower associated pressure drop, leading to lower operational costs. The inlet diameter also showed a significant influence on the pressure drop within the system.The sensitivity analysis on membrane properties and operating conditions revealed that the inlet pressure, membrane permeability, membrane thickness and membrane area have a significant impact on the total permeate flux.The membrane area size provided useful information for optimization purposes: by reducing the membrane area to the size of the spacer, more, smaller membranes can be combined to obtain a higher flux.

### Future Work

To validate the simulation model and optimize the module design, experimental data of the pressure drop in the retentate or permeate channels with a spacer needs to be determined. This is because in module and plant design and optimization studies, permeate pressure drop is a design variable which cannot be ignored. Concentration polarization (CP) is another important factor that limits separation performance in nearly all membrane filtration processes. Thus, accurate prediction of CP and fouling phenomena is also critical for design and optimization processes.

## Figures and Tables

**Figure 1 membranes-11-00054-f001:**
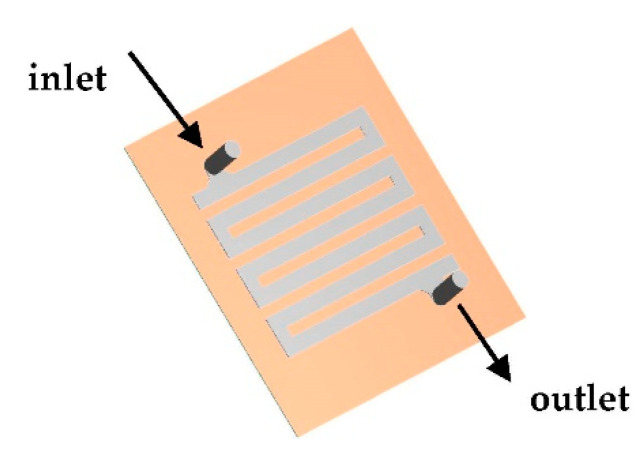
Top-down view of the simulated geometry. Top side is the inlet and bottom side is the outlet.

**Figure 2 membranes-11-00054-f002:**
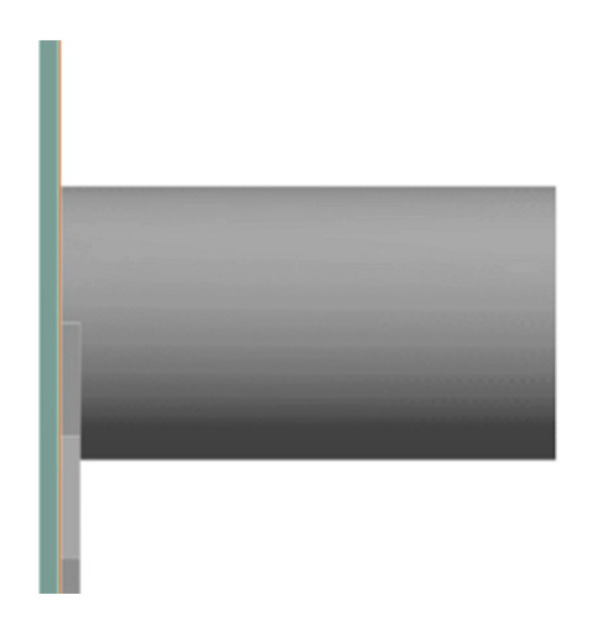
Side view of a part of the simulated geometry near the inlet. Visible are inlet of the inflow zone (grey), the very thin membrane zone (orange) and the outflow zone (green).

**Figure 3 membranes-11-00054-f003:**
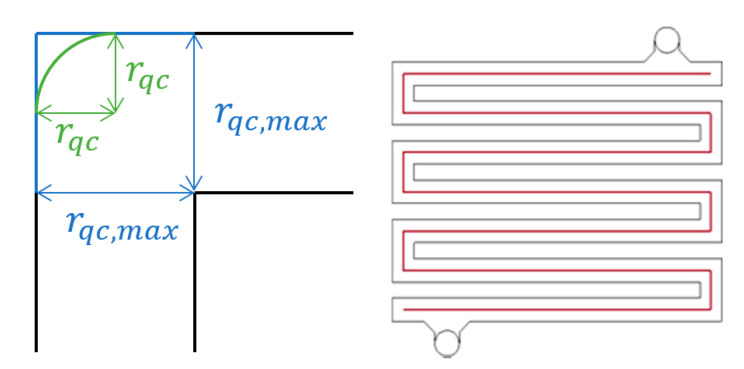
Graphical overview to how the “curviness” parameter (*r_qc_*—green and *r_qc,max_*—blue) is calculated (left) at the different corners of the spacer geometry (right). The fluid flow path within the spacer is shown in red.

**Figure 4 membranes-11-00054-f004:**
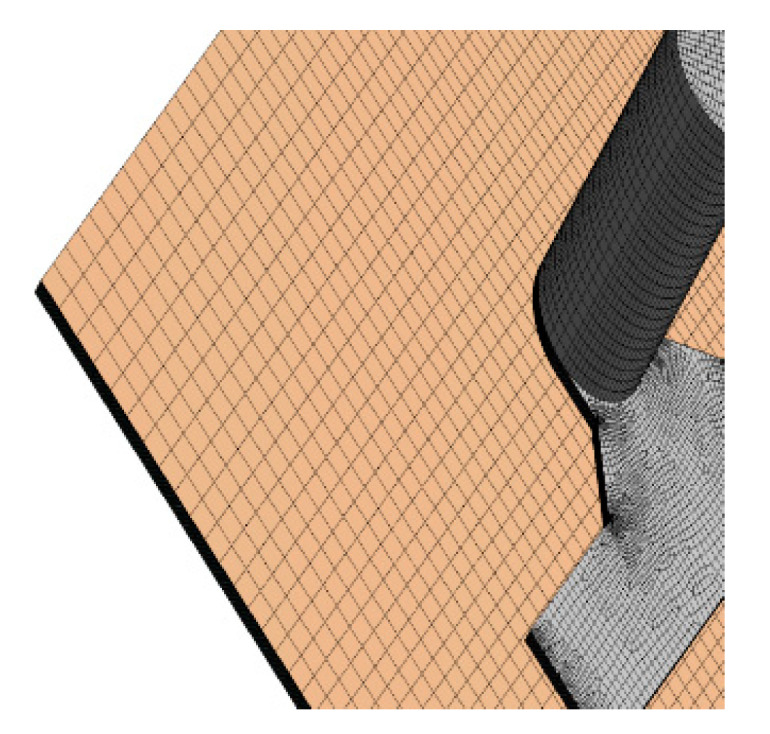
Close-up view of the structured multigrid. Below the membrane is the outflow zone, meshed finely in the direction of the flow.

**Figure 5 membranes-11-00054-f005:**
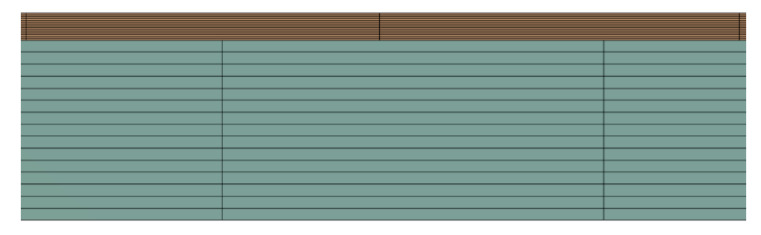
Zoomed-in side view of the membrane zone (orange) and the outflow zone (grey) grid.

**Figure 6 membranes-11-00054-f006:**
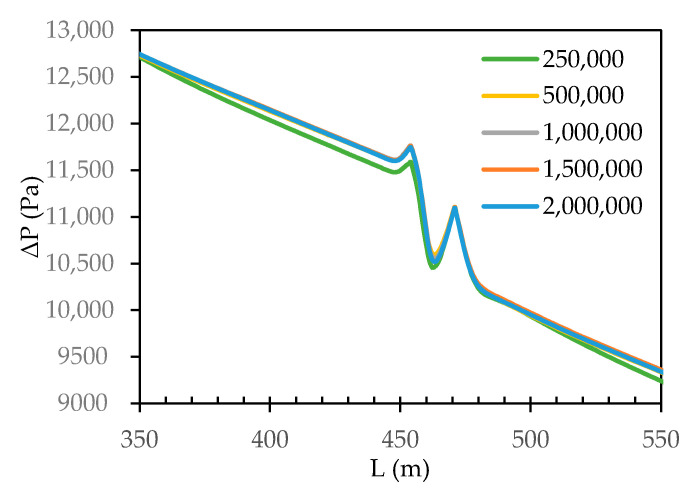
Development of the transmembrane pressure profile along the centreline in the flow path for the five grid cell counts.

**Figure 7 membranes-11-00054-f007:**
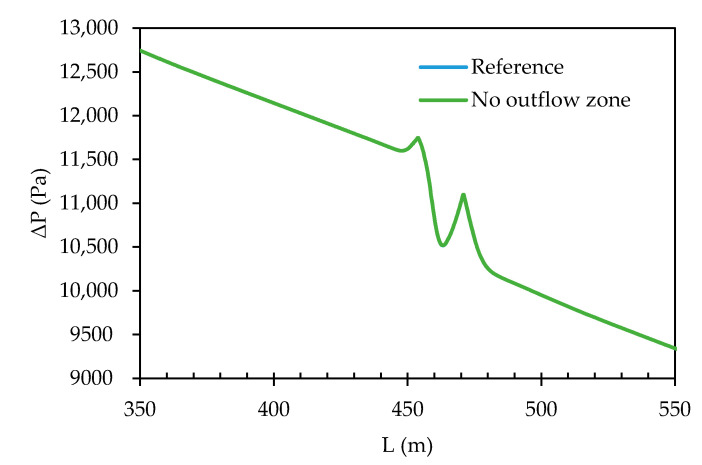
Development of TMP along the spacer flow path centreline for the 1,500,000 cell grid with an outflow zone (reference) and without an outflow zone.

**Figure 8 membranes-11-00054-f008:**
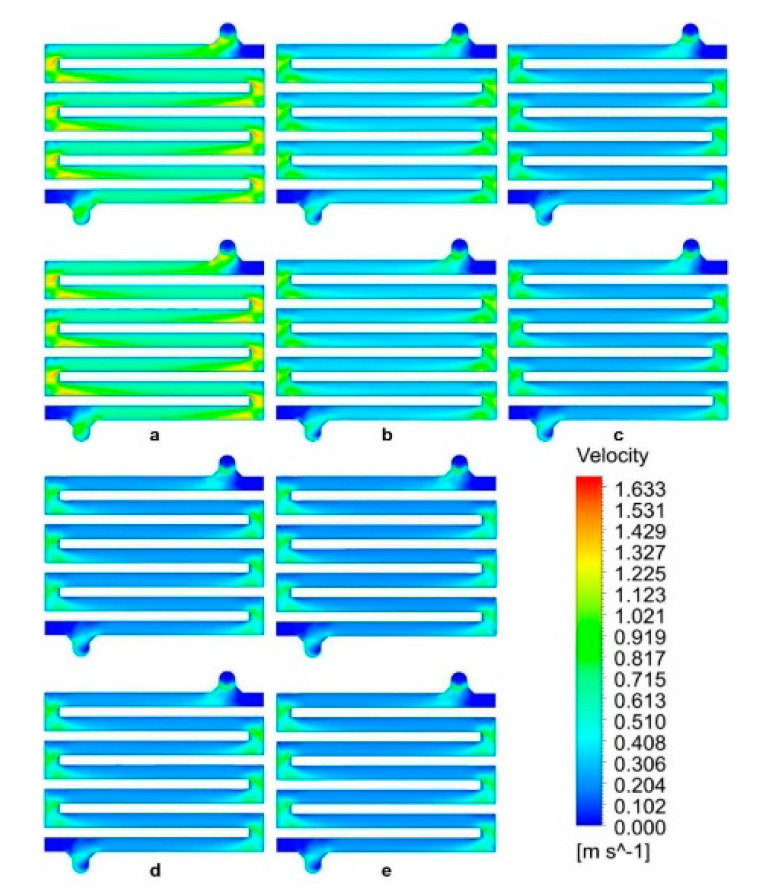
Velocity contour plots for the inflow path through the spacer next to the membrane, for cases with an outflow zone (upper figures) and without an outflow zone (lower figures) grids; (**a**) 250,000 cells, (**b**) 500,000 cells, (**c**) 1,000,000 cells, (**d**) 1,500,000 cells and (**e**) 2,000,000 cells.

**Figure 9 membranes-11-00054-f009:**
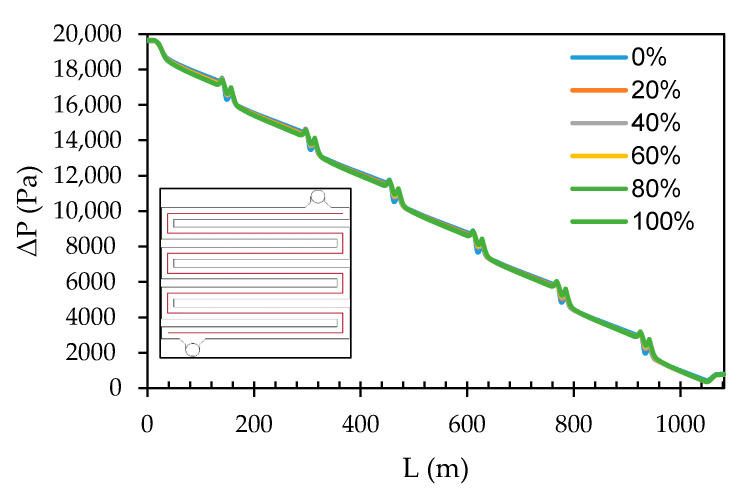
Development of TMP along the centerline in the flow path at different spacer curviness values.

**Figure 10 membranes-11-00054-f010:**
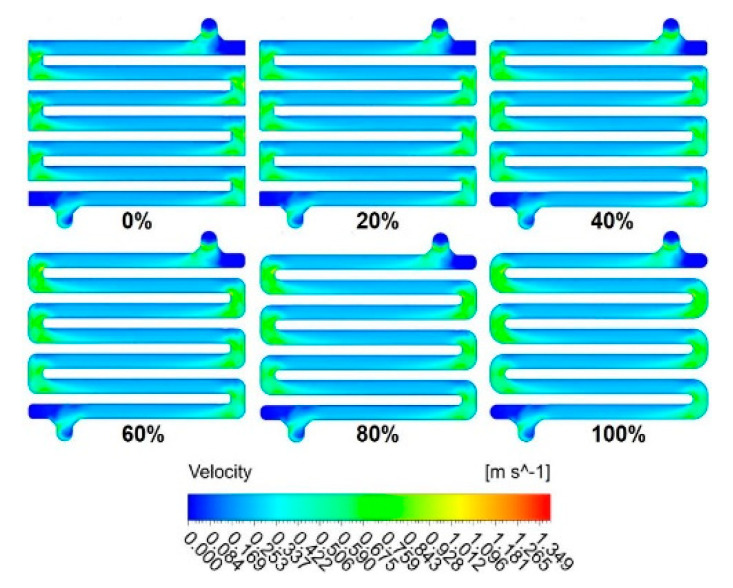
Velocity contour plots for the inflow through the spacer next to the membrane for all curviness cases: (**a**) 0%, (**b**) 20%, (**c**) 40%, (**d**) 60%, (**e**) 80%, and (**f**) 100%.

**Figure 11 membranes-11-00054-f011:**
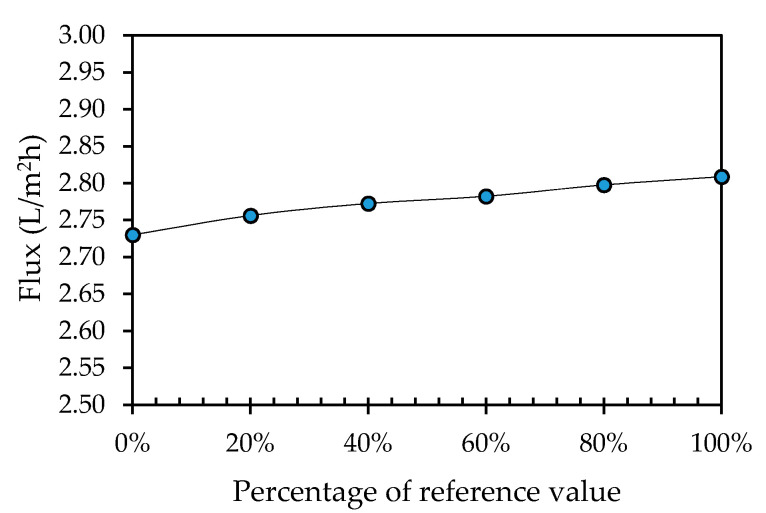
Total permeate flux leaving the module plotted as a function of the percentage of the reference value (100% corresponds to 0% curviness, 80% to 20% curviness, 60% to 40% curviness, 40% to 60% curviness, 20% to 40% curviness).

**Figure 12 membranes-11-00054-f012:**
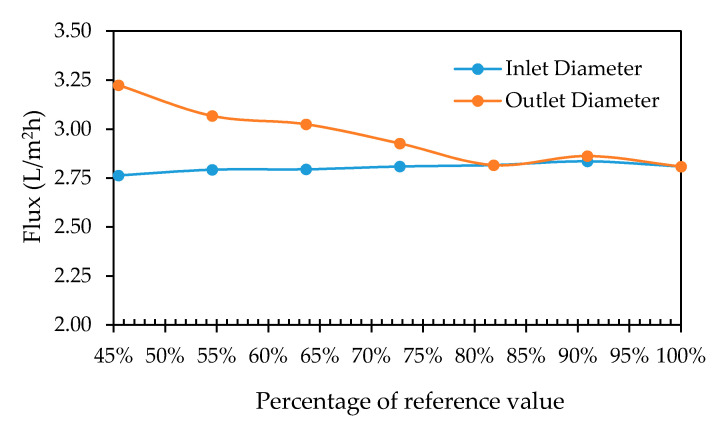
Total permeate flux leaving the module plotted as a function of the percentage of the reference value.

**Figure 13 membranes-11-00054-f013:**
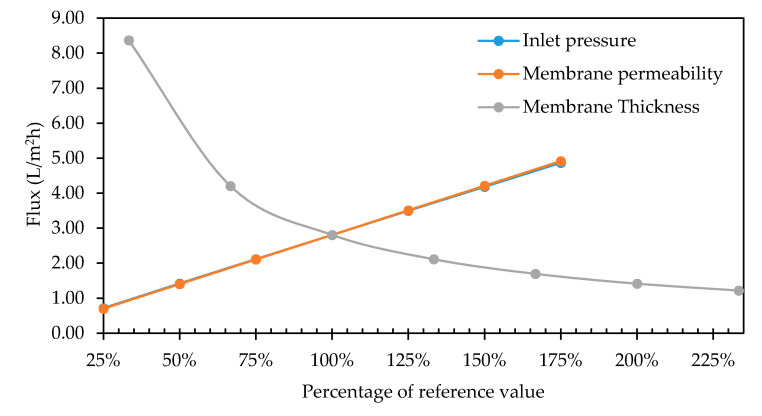
Relationship between intrinsic membrane permeability, total pure water flux leaving the module and membrane thickness.

**Figure 14 membranes-11-00054-f014:**
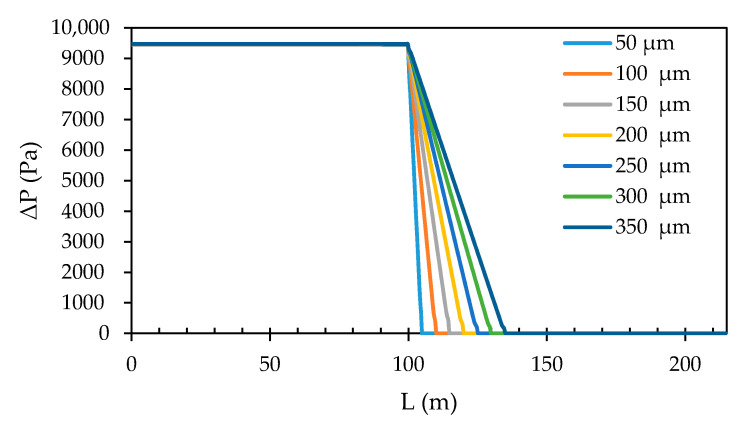
Development of the pressure profile through the spacer zone, the membrane zone and the outflow zone at different membrane thickness values.

**Figure 15 membranes-11-00054-f015:**
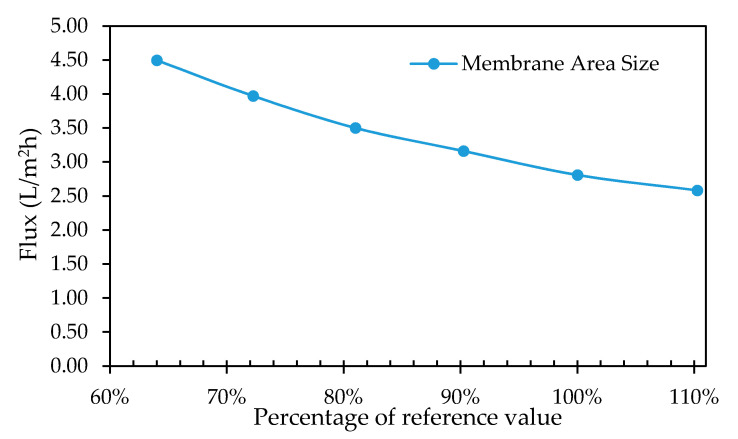
Total permeate flux leaving the module plotted as a function of the percentage of the reference value for surface area.

**Table 1 membranes-11-00054-t001:** Overview of operational parameters values used in a sensitivity analysis.

Parameter	Value						
Intrinsic Permeability (10^−17^ m^2^)	1.042	2.084	3.125	4.167 ^1^	5.209	6.251	7.292
Membrane thickness (µm)	50	100	150 ^1^	200	250	300	350
Inlet pressure (bar)	0.05	0.10	0.15	0.20 ^1^	0.25	0.30	0.35

^1^ Values for the reference case.

**Table 2 membranes-11-00054-t002:** Overview of geometrical parameters values used in a sensitivity analysis.

Parameter	Value						
Spacer thickness (mm)	0.5	1 ^2^	1.5	2.0	2.5	3.0	
Spacer curviness (%)	0 ^2^	20	40	60	80	100	
Inlet length (mm)	10	15	20	25 ^2^	30	35	40
Outlet length (mm)	10	15	20	25 ^2^	30	35	40
Inlet diameter (mm)	5	6	7	8	9	10	11 ^2^
Outlet diameter (mm)	5	6	7	8	9	10	11 ^2^

^2^ Values for the reference case.

**Table 3 membranes-11-00054-t003:** Qualitative overview of several sensitivity analysis parameters on the permeate flux and the transmembrane pressure.

Parameter	Permeate Flux	Transmembrane Pressure
Spacer curviness	↓	↓
Thickness of the spacer	↓↓	↓↓
Thickness of the outflow zone	–	–
Length of the inlet pipe	–	–
Length of the outlet pipe	–	–
Diameter of the inlet pipe	↑	↑
Diameter of the outlet pipe	↓↓↓	↓↓↓
Inlet pressure	↑↑↑↑	↑↑↑↑
Membrane permeability	↑↑↑↑	↓
Membrane thickness	↓↓↓↓	↑
Membrane area size	↓↓↓↓	–

↑: increase of either the flux or the transmembrane pressure with an increase of the parameter; ↓: decrease; –: no clear dependency between either the flux or the transmembrane pressure and the sensitivity analysis parameter.

## Data Availability

Not applicable.

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
