# Peer review of "Numerical Modelling Assisted Design of a Compact Ultrafiltration (UF) Flat Sheet Membrane Module"

_membranes, 2021, doi:10.3390/membranes11010054_

Round 1

Reviewer 1 Report

The manuscript presents a parametric CFD study for the design of a flat sheet UF membrane system. Only hydrodynamics and pressure profiles are simulated, which are used to predict flux profiles along the membrane channel. Emphasis is made on the mesh independence study and selection of a turbulence model, before conclusions are presented regarding the effect of each of the different input parameters varied, which include the length and size of the inlet and outlet, the corner geometry, membrane and spacer thickness, as well as membrane permeance and inlet pressure.

The manuscript is generally well written and does not require major English language editing. However, it first presents a series of results, followed by a discussion section. This awkward presentation order, combined with the manuscript referring to several figures in the supplementary material, make the reader have to go back and forth between pages and documents to understand the point the authors are trying to make. Furthermore, the “discussion” is more of a description and interpretation of the results, with very few insights into the implications or relevance of these in terms of membrane module design or membrane performance improvements.

Overall, the manuscript reads more like a case study report than a scientific paper. I therefore cannot recommend it for publication in its current form, as it also has some other specific issues that I list below:

  • The introduction mentions as the study’s aim the design and optimisation of the UF membrane module. However, the conclusions seem to be mostly focused on the mesh independence for this particular design. If the module is to be optimised using CFD, mesh independence would need to be carried out for each additional design option. Hence, emphasising the mesh independence study does not seem to fit well with the objective of this study. The mesh independence section should be reduced in size and emphasis, and only provide enough data and evidence to guarantee the reader that no externalities are introduced by the meshing scheme (e.g. report the Grid Convergence Index for each mesh).
  • The introduction mentions that fouling and concentration polarisation are two of the main problems faced by these type of membrane systems. However, these two phenomena are not included in the simulation. The authors should explain how the module could be considered “optimised” if both these issues are neither addressed nor considered.
  • Several mentions to the application of these UF systems in small rural communities are made in the introduction, but it is unclear how that is relevant. What makes the application of these membrane units significantly different in those communities?
  • The geometry proposed for the analysis is very different to most commercial systems used in industry. In particular, the geometry used in the study sacrifices large portions of membrane area. How would this geometry be used in a plate and frame system?
  • The authors should provide more information in the geometry figures, particularly indicating the direction of fluid flow, where the inlet and outlet are located.
  • Even though there is a lot of emphasis on the mesh independence study, there is significant lack of clarity about the mesh used. An example mesh should be provided for the benefit of the reader. How is the mesh refined? Are elements incremented everywhere or only in particular locations. How are the rounded corners handled if hexahedral cells are used? Is the mesh structured or unstructured? Are inflation regions used for the solid surfaces? If not, why not?
  • Figure 3 should define what each colour represents.
  • Equation 2 is only the mass conservation equation. The general momentum conservation equation (not for an incompressible fluid) is not presented.
  • Equation 3 has parentheses around the whole equation, these need to be removed.
  • The authors mention that they assume constant density, yet the density in the first term in equation 4 appears inside a time derivative. This can be taken out and perhaps the whole equation divided by density.
  • On page 7, the authors make reference to the permeability “K”. However, the units of this property mean this is actually “permeance” not permeability. Permeability is a property of the membrane itself, unaffected by its thickness, whereas permeance is permeability divided by thickness (the thicker the membrane the lower the permeance as there is more resistance to fluid flow). These concepts are related to resistance vs. resistivity, which are the inverses of permeance and permeability respectively.
  • The section describing the turbulence models, from the bottom of page 7 (around equation 7) through to just before the last paragraph before section 3, is unnecessary and should be summarised and removed. Interested readers can be referred to the many textbooks or other publications that give more details about each turbulence model.
  • In page 8, the authors suggest that the velocity in the module depends on the inlet pressure. However, pressure and velocity can be independently controlled in a physical module via a back-pressure valve. Are the authors assuming atmospheric pressure at the module outlet? This is not typical operation for UF systems, which can be in many cases dead-end units. Nonetheless, cross-flow units typically incorporate a back-pressure valve to ensure a minimum pressure throughout the module. This needs to be clarified.
  • On page 9, references to Figures S1 and S3 seem to be reversed.
  • Figures 4 and 5 have very small font and are quite blurry. Higher quality images need to be provided.
  • Section 3.3.4 heading refers to changes in diameter, but the text mentions changes to inlet and outlet length.
  • It is unclear how the hydraulic diameter is calculated for determining the Reynolds number
  • The section on turbulence modelling seems unnecessary, as the authors conclude that the fluid flow regime is laminar in the end. It is unclear why the authors considered using a turbulence model in the first place. Much literature in CFD modelling of membrane system is clear in that flow in most of these systems is laminar in nature, and perhaps unsteady (transient).
  • Much of the discussion section seems to rationalise the results on the hypothesis that water tends to “accumulate” in the corners of the geometry. However, I fail to understand how water can accumulate in this type of system, when it is incompressible and density is constant. The authors should attempt to analyse their results and interpret them on well-established fluid-dynamics theory, e.g. pressure losses due to form and skin drag.

It is questionable whether there are any significant contributions to the understanding of either CFD modelling of membrane systems or to the impact of hydrodynamics on the performance of this particular geometry. As such, I cannot recommend this manuscript for publication.

Reviewer 2 Report

Review MDPI  n.   994443

The manuscript deals with an interesting topic.

A wide and clear introduction leads the reader to understand the problem and to appreciate the novelty of gravity-driven UF prototypes.

The methodology of investigation is clear: a systematic study is carried out with the tools of the CFD to investigate the effect of geometrical parameters (spacer geometry, cell geometry) as well as of membrane parameters and of inlet pressure.

I appreciated the collection of the numerous results in the Supplementary material.

I suggest publication after minor revision

Some detailed comments

I found no clear information about the permeate spacer: Could you insert please a comment in the paragraph 2.1?  Is the “outflow thickness” the thickness of the permeate side?

In Table 2, “inlet pressure(bar)” is reported. I wonder if it is the inlet pressure in barg or if it is the TPM at the inlet section. Could you clarify please?  If a permeate spacer is present, it can make the difference, owing to pressure drops in the permeate side.

Some conclusions are rather obvious. Sections 4.3.1, 4.3.2 and 4.3.3 are presented as important results, whereas they are only a reconfirmation of the calculation procedures. The linear increase of the flux with TMP is intrinsically contained in Eq.(5) (!.).

Comments about the effect of the membrane area (paragraph 4.3.4) are not clear to me. Which parameters are varied in figures S13 and S14, in addition to the membrane area ?  Which parameters are kept constant?  It would be useful to show in the caption which are the corresponding cases of Table 1 and 2 for the calculations, for instance.
